# A Comparison of Internal, Marginal, and Incisal Gaps in Zirconia Laminates Fabricated Using Subtractive Manufacturing and 3D Printing Methods

**DOI:** 10.3390/biomimetics9120728

**Published:** 2024-11-28

**Authors:** Mijun Noh, Jaehong Kim

**Affiliations:** 1Department of Healthcare Sciences, Faculty of Dental Laboratory Science and Engineering, Korea University, 145 Anam-ro, Seongbuk-gu, Seoul 02841, Republic of Korea; mijune97@korea.ac.kr; 2Department of Dental Laboratory Science, College of Health Science, Catholic University of Pusan, 57 Oryundae-ro, Geumjeong-gu, Busan 46252, Republic of Korea

**Keywords:** 3D printing, digital light processing, computer-aided-design/computer-aided-manufacturing, subtractive manufacturing, zirconia

## Abstract

DLP printing is a new method for producing zirconia laminates that ensure clinically acceptable gaps in the internal, marginal, and incisal regions. A typical model of a central maxillary incisor was prepped by a dentist and scanned. The laminate was designed using CAD software version 2023. The laminates were fabricated using a milling machine (LSM group) and a DLP printer (LAM group) (N = 20). The gap was evaluated using the silicone replica method at designated measurement points. Statistical analyses were performed. The Shapiro–Wilk and Kolmogorov–Smirnov tests indicated a non-normal distribution, and the Mann–Whitney test was used. The LSM group had wider gaps than the LAM group except at point E (59.5 µm). The LAM group had wider gaps than the LSM group, except at points H (51.70 µm). No significant differences were observed between the LSM and LAM groups at any of the labiolingual measurement points. In the mesiodistal plane, a significant difference was observed between the two groups at point G, which was adjacent to the mesial side (*p* < 0.05). The results of this study indicate that DLP printing offers an innovative approach for producing zirconia laminates, as the incisal, internal, and marginal gaps are within clinically acceptable ranges compared with the AM method.

## 1. Introduction

Currently, the purpose of dental prosthetic treatment has evolved beyond merely re-storing caries to include restoring the form and color of the teeth [1,2]. Restorative dentistry is closely related to human well-being, and extensive research is being conducted to enhance the mechanical properties and durability of dental restorative materials [3]. As patients’ interest in aesthetics has increased, the demand for aesthetic prosthetics has in-creased [4]. Aesthetic prosthetics is a challenging field that must consider harmony with the patient’s facial features, harmony between teeth and periodontal tissues, and continuity with the surrounding teeth [5,6,7]. Aesthetic prostheses are made of materials with optical properties similar to those of natural teeth, such as glass-based lithium disilicate ceramics, alumina, and zirconia [8]. Biocompatible materials that are aesthetically pleasing and do not irritate the periodontal tissues are suitable for aesthetic prosthetics [9,10]. Therefore, prosthetics made from monolithic zirconia have emerged as materials for aesthetic prosthetics because of their optical properties similar to those of natural teeth, high strength, and biocompatibility [11,12,13].

The mainstream method for fabricating monolithic zirconia restorations involves the subtractive manufacturing (SM) method using pre-sintered zirconia blocks [14]. However, the subtractive method has drawbacks, including possible adverse effects due to microcracks occurring during the milling process, material wasting, inability to reuse mate-rials, and difficulties in shape geometry formation because of the size of the milling tools and axes of the milling machine [15,16,17]. Currently, a wide range of dental treatment trials, including orthodontics, dental implants, mandibular reconstructions, prosthodontic re-habilitation, and surgical and nonsurgical endodontics, have extensively exploited additive manufacturing (AM) methods to overcome these issues [18,19,20,21,22].

Among AM methods, stereolithography (SLA), digital light processing (DLP), and lithography-based ceramic manufacturing are currently used to fabricate monolithic zirconia restorations [23]. The principles of DLP and SLA are similar, with the difference being the choice of light source [24]. Stereolithography (SLA) employs a laser system, whereas DLP uses a digital micromirror device (DMD) to reflect a projected ultraviolet (UV) light source through a rectangular array of micromirrors. In the DLP technology, the UV light source rapidly cures the photopolymer material, and the DMD, with its array of microscopic mirrors, digitally patterns light by tilting each micromirror individually. This allows DLP to cure an entire layer at once, achieving a higher production speed than SLA by reducing the layer-by-layer manufacturing time and enhancing the production efficiency [25]. The parameters of the light source can also be adjusted to improve the production quality.

These methods involve photopolymerizing a slurry/paste that combines a photocurable binder with zirconia particles to create a green body, which then undergoes thermal processing such as debinding and sintering. This process was required because zirconia has a higher melting point than the binder (resin), rendering direct fusion inefficient in terms of time and energy [26]. The debinding process removes the binder from the green body, which is crucial for ensuring the integrity of the final product. Generally, this is achieved using thermal, solvent, or catalytic methods [27,28,29,30]. Thermal debinding in-volves heating a green body to the temperature at which the binder evaporates or degrades [31]. Proper temperature control is essential to avoid defects such as cracking and warping [32]. After debinding, the zirconia parts undergo sintering, which densifies the zirconia by heating it to a temperature below its melting point, allowing the ceramic particles to bond. This process eliminates porosity and increases the mechanical strength of zirconia [33]. Sintering temperatures for zirconia are typically very high, around 1400–1600 °C, and precise control of the heating and cooling rates is necessary to achieve optimal density and mechanical properties [34]. The characteristics of DLP technology ensure high production speed and quality, making it particularly advantageous for creating com-plex-shaped monolithic zirconia prostheses [35].

From a clinical perspective, the fit of a prosthesis is the gap between the inner and outer surfaces of the abutment tooth [36]. This is termed a fit or gap; the closer it is to zero, the higher the quality and suitability of the fit. As with all restorations, fit considerably impacts the longevity of the laminates [37]. A poor fit can lead to cement dissolution, causing decay and periodontal diseases as well as problems such as prosthesis fracture and dislodgement [38]. For laminates produced using the computer-aided design/computer-aided manufacturing (CAD/CAM) method, which is a subtractive method, any misfit at the laminate margin should be <120 µm [39]. Reportedly, when CAD/CAM is used, this restriction should not be >100 µm [40].

Therefore, evaluating the fit of prostheses fabricated using technologies with limited clinical adoption is crucial. The fit of AM zirconia laminates has not been sufficiently studied compared to that of prostheses fabricated using CAD/CAM [39,40]. Numerous studies have been conducted on the fit of 3D-printed monolithic zirconia prostheses. However, studies on the internal and peripheral fits of zirconia laminates manufactured by 3D printing are lacking. In dentistry, the ceramic DLP method is currently in the introductory phase, and active research is ongoing to enable its clinical applications. This study also serves as a foundation for the clinical application of ceramic DLP methods in the fabrication of laminates. Studies have only been conducted on the fit of laminates manufactured using only SLA 3D printing [41,42]. Thus, this study aimed to investigate whether DLP printing can produce precise laminates compared to the SM method. The null hypothesis of this study is that “there is no difference between the fit of zirconia laminates produced by the DLP printing and milling methods”.

## 2. Materials and Methods

A typical model with a central maxillary incisor (D85DP-500B.1; Nissin Dental, Tokyo, Japan) was selected as the study model and prepared by a dentist in a butt joint incisal preparation design. The prepared abutment die was scanned using a dental scanner (E4, 3Shape A/S, Copenhagen, Denmark), and anatomically shaped zirconia laminates were designed as thin as 0.3 mm using CAD software (3Shape Dental Designer version 2023, 3Shape A/S, Copenhagen, Denmark) and saved as an STL file (Figure 1). The images were exported to the respective CAM software packages for printing and milling.

The specimens manufactured using the subtractive manufacturing method were designated as the LSM group (*n* = 10). First, the tool paths were calculated from the STL files in the CAM software (HyperDENT V9.2, FOLLOW-ME Technology Group, Berlin, Germany). The pre-sintered 3 mol% zirconia disk block (Luxen Zirconia 1200 Zr, Dntalmax Co., Seoul, Republic of Korea) was cut using a 5-axis milling machine (K5+; VHF, Berlin, Germany). The supporters of the milled specimens were removed and sintered using a sintering furnace (EX-6100, Add-in, Seoul, Republic of Korea) at a maximum temperature of 1550 °C according to the block manufacturer’s recommended schedule. No additional operations such as polishing were required (Figure 2).

Specimens manufactured using the additive manufacturing method were designated as the LAM group (*n* = 10). The LAM group positioned the STL file on the build platform using the dedicated slicing software (ZIPROS, Aon, Seoul, Republic of Korea) of the 3D printer [43]. Support was provided and stored in accordance with the manufacturer’s instructions. The bath of the 3D printer was filled with a zirconia paste (ININI-CERA, Aon, Seoul, Republic of Korea). The paste consists of 3-mol%-yttria-stabilized zirconia powder with a photopolymerized resin serving as the binder. The output layer height was set to 25 μm. After removing the support structures from the output, the residual zirconia paste was removed using a brush and alcohol solution, and both the internal and external residues were cleaned for 2 min in an ultrasonic cleaner filled with an isopropanol alcohol solution. The debinding process was conducted using a sintering machine (CERAFUR, INWHAHNC, Seoul, Republic of Korea) by increasing the temperature at a rate of 0.2 °C per min for 1 h and 0.5 °C per min for another hour until it reached 500 °C. The specimens were then sintered by increasing the temperature from 500 °C to a maximum of 1500 °C and maintaining the temperature for 2 h. No additional operations such as polishing were required (Figure 2).

A gap in a prosthesis is the discrepancy between the inner and outer surfaces of an abutment tooth (Figure 3). This gap was measured using the silicone replica method with a fit-checking silicone material (Figure 4). The abutment die was printed using a DLP 3D printer (Asiga, Sydney, Australia), and a liquid resin for the dental model (Asiga DentaModel, Asiga, Sydney, Australia) for the silicone replica method. A layer height was set at 25 μm. The die was cleaned for 10 min in an ultrasonic cleaner filled with an isopropyl alcohol solution and cured using a UV curing machine (CureM, Graphy, Seoul, Republic of Korea) for 10 min, according to the manufacturer’s instructions.

Abutment die fitting was performed by injecting a fit-checking silicone material (FIT CHECKER ADVANCED, GC Corporation, Tokyo, Japan) into the inner surface of the zirconia laminate [44,45]. Before injection, equal lengths of the base and catalyst were squeezed onto the mixing pad and mixed thoroughly for 20 s. The laminates were then placed directly on a die. After 1 min of polymerization with a 20-N vertical compression force using a universal testing machine (Instron 3345, Instron Corporation, Canton, OH, USA), the specimen was removed, retaining the silicone film that represented the gap outside the abutment. This film was then fixed using heavy-body silicone (hydrophilic vinyl polysiloxane impression material; Spident, Seoul, Republic of Korea) and light-body silicone (Aquasil Ultra XLV Regular Set; Dentsply Sirona, NY, USA). Afterwards, it was cut with a number 11 surgical scalpel blade (Henry Schien Inc., Melville, NY, USA), according to the sectioned planes. Ten measurement points were assigned to each plane (Figure 5). Points A, B, C, D, and E were cut labiolingually. Measurement points F, G, H, I, and J were cut mesiodistally. Finally, the gaps were measured using a digital microscope (KH-7700; Hirox, Tokyo, Japan) at a 160× magnification (Figure 6).

Statistical analysis of the measurements was conducted using Statistical Package for Social Sciences version 23 (IBM Corp., Armonk, NY, USA). To test the regularity of the fit in the measured areas, the Shapiro–Wilk and Kolmogorov–Smirnov tests were conducted, and the results did not indicate a normal distribution (*p* < 0.05). Therefore, the Mann–Whitney test was used.

## 3. Results

The fit of the zirconia laminate measured in the labiolingual and mesiodistal planes is presented in Table 1 and Table 2, respectively. The measurement points for the labiolingual planes were the marginal (A, E), internal (B, D), and incisal (C). The measurement points for the mesiodistal planes were the incisal planes (F, G, H, I, and J). No significant differences were observed between the LSM and LAM groups at any of the labiolingual measurement points. Only at point E, corresponding to the labial margin, the LSM group exhibited a gap of 56.2 μm, while the LAM group had a wider gap of 59.5 μm, with LSM showing wider gaps at all other points. Additionally, the widest gap was observed at point C for LSM, measuring 79.7 μm, while the narrowest gap was at point E for LSM, measuring 56.2 μm.

In the mesiodistal planes, a significant difference was observed between the two groups at measurement point G, which was adjacent to the mesial side (*p* = 0.004). However, no significant differences were observed at the measurement points F, H, I, and J. The widest gap was observed at measuring point I for the LAM group, measuring 53.50 μm, while the narrowest gap was at measuring point G for the LSM group, measuring 40.70 μm. Box plots of the LSM and LAM groups are shown in Figure 7.

## 4. Discussion

In this study, the gap between the zirconia laminates fabricated by the control group using the CAD/CAM method and the test group using the DLP printing method was evaluated using the silicone replica method at designated measurement points in the labiolingual and mesiodistal planes. The null hypothesis was that “there is no difference between the groups at all measurement points for laminates produced using CAD/CAM and DLP printing methods”. The results of this study showed no significant intergroup differences at points A, B, C, D, and E in the labiolingual plane. In the mesiodistal plane, a significant difference was observed at measurement point G (*p* < 0.05) and no significant difference at measurement points F, G, H, I, and J. Thus, the null hypothesis was not rejected in the labiolingual plane, but in the mesiodistal plane.

The accuracy of the marginal gap generally depends on factors such as tooth preparation, impression techniques, materials and technologies used in prosthesis fabrication, and adhesive cement [46,47,48,49]. Various methods have been developed and refined to measure the fit of prostheses accurately. These measurement methods include cross-sectional measurement, the silicone replica technique, micro-computed tomography (CT), and 3D overlay evaluation [50,51,52,53,54]. The cross-sectional measurement method involved cementing the prosthesis with the prepared tooth, cutting a section along the desired direction, and measuring it using a microscope. The silicone replica technique involves injecting silicone into the inner surface of the prosthesis, allowing it to harden, and cutting the silicone im-pression to measure the gaps using a microscope. This method allows specification of the direction of the cross section as intended by the surgeon, thereby providing a wide measurement range. Another method involves using a 3D scanner to scan both the silicone im-pression and prosthesis and then overlaying their 3D images to measure the internal gaps. The Micro CT method uses radiographic imaging to obtain high-resolution 2D and 3D images of the prosthesis, and the gaps are measured. The software automatically performed measurements and calculated the required values. Finally, the 3D overlay evaluation method sets the STL file designed using the CAD software as a reference file. It scans the prosthesis to be evaluated using a dental model scanner and measures the volumetric gap by superimposing the scan data obtained using the Root Mean Square (RMS) value.

Among these methods, this study employed the silicone replica technique, which is commonly used and is accurate [55]. The silicone replica technique measures the fit or cement space of a prosthesis without causing damage. In clinical terms, prosthesis fit refers to the gap between the inner surface of the prosthesis and outer surface of the abutment. Therefore, internal fit represents prosthesis accuracy [56]. Because direct measurement of the internal fit of prostheses, which usually have complex 3D structures, is challenging, this gap has primarily been measured using the replica method with an elastomeric material [57,58]. However, the silicone replica method has some limitations. First, the risk of silicone tearing may occur during the process of separating the abutment and prosthesis; therefore, the skill of the fitting technician affects data acquisition. Second, errors may occur during the scanning process used to obtain the 3D data to create the abutment for fitting. Third, light-body silicone, which serves as a gap during the cementation process, may cause expansion and contraction, leading to volume changes and potentially inaccurate measurements. To control these issues, this study employed a fit-checking silicone material that minimizes shrinkage and expansion by increasing the hardness after polymerization compared with light-body silicone. This material was used before cementation to accurately check the fit of the prosthesis in clinical practice. Finally, compared to methods that measure gaps based on accurate location information using digital technology, the consistency and accuracy of the measurement location are low [52,53]. This is a limitation of the present study.

Previous studies have shown that anterior veneers have a minimum thickness of 0.3–1.5 mm and the mean marginal gap varied between 125 and 402 µm at the incisal margin; moreover, 100–150 µm has also been recommended by several authors as clinically acceptable with regard to longevity [59,60,61,62,63]. A scanning electron microscope was used to measure the conformity of milled laminates, and previous studies have shown that the marginal gap at the incisal edge was 203–289 µm, and the marginal gap at the cervical edge and the internal gap were 87–147 µm and 71–104 µm, respectively [64]. The results of this study show that average gaps at measurement points (40.7–79.7 µm) are smaller than that clinically acceptable with regard to longevity suggested by previous studies.

In a recent study that measured the gap of zirconia laminates fabricated using DLP 3D printing with Micro CT, the mean and maximum marginal gaps were 55 ± 9 µm and 143 µm, respectively, and for the labial margin, they were 68 ± 14 µm and 130 µm, respectively [65]. In this study, the means of A and E, the measurement points corresponding to the margin of the PT group, were 59.1 and 59.5 µm, respectively, which were higher but similar to the previous study’s values, with maximum gaps of 80 and 72 µm, respectively, both of which were lower than those in a previous study. The range of gaps appeared to be relatively similar, and factors such as the shape of the abutment margin preparation, composition of the zirconia paste, printer settings, scanning errors, and measurement method errors may have affected the fit [66,67,68,69].

This study found no significant differences at any point except for point G, indicating that the 3D printing method has the potential to produce zirconia laminates with gaps as accurately as those produced using the CAD/CAM method. However, previous studies have shown significant differences in the marginal and labial area gaps between milled and 3D-printed veneers, resulting in contrasting findings [65]. In this study, an ultrasonic bath cleaning method was used to remove slurry residues from the sintered green body, whereas previous studies used an isopropanol spray for air brushing. The process of thoroughly removing the residual resin binder without delaminating the layers of the green body after zirconia printing is essential because the cleaning method can affect the wear, biocompatibility, and mechanical properties [70].

Isopropanol is generally recommended to dissolve resin components when cleaning polymer-based 3D-printed outputs, and an ultrasonic bath cleaning method is typically employed. According to previous studies that examined the effects of air brushing with isopropanol and ultrasonic bath cleaning on the geometry, transmission, roughness parameters, and flexural strength of 3D-printed zirconia, air brushing was superior in terms of transmission, roughness, and strength. In addition, ultrasonic cleaning results in the greatest geometric inconsistency because of the residual resin binder [71]. These studies used disk-shaped samples, which limited their applicability to hollow structures such as prostheses. However, further research is necessary to confirm whether the cleaning process affects the final fit of 3D-printed prostheses.

In zirconia-paste-based photopolymerization printing, supports play a crucial role in directly stabilizing the prosthesis on the 3D printer plate. Current research is ongoing on the development of non-contact supports, volumetric accuracy according to the shape and number of supports, and the presence or absence of auxiliary supports. In this study, debinding and sintering were performed after removing the support from the specimens, according to the manufacturer’s recommendations. Previous studies have indicated that by removing supports immediately after green-body printing, destructive damage is caused to the surface of the green body during the removal process, which is in direct contact with the prosthesis. After debinding and sintering, zirconia becomes exceptionally hard and brittle, resulting in low processability. When the supports were removed via hard milling, various cracks were formed on the surface of the prosthesis. Consequently, the milling method that cuts the pre-sintered zirconia block also removes the support structure by grinding milling before sintering. Therefore, an accurate analysis of the effect of the support removal timing on the dimensional accuracy of the output is essential.

Laminates primarily utilize esthetic ceramic materials with translucence, which can replace dentin [72]. In this study, a white paste containing 3 mol% zirconia was used. To address this issue of lacking esthetics, development efforts have focused on zirconia pastes [73,74,75]. However, integrating these materials with a 3D printing process, which is noted for its high manufacturing precision in producing thin thicknesses and sharp margins, presents a promising alternative [64]. This approach aims to surpass the conventional manufacturing methods caused by errors in wax pattern production, investment, pressing, and operator mishaps [76]. The efficacy of zirconia in the manufacture of esthetic prostheses was assessed by comparing it to the milling method, which is the established gold standard. Gap measurements were conducted at internal and marginal cutting points to verify the utility of this innovative manufacturing technique. In addition, the gap was measured using the silicone replica method, which requires fewer measurement points than the CT method, thereby limiting the overall measurement range. As the 3D-printed zirconia green body undergoes deformation or volume changes during cleaning, debinding, and shrinkage during the sintering process, the probability of errors occurring at various stages of the process is high [77]. A follow-up study with a larger sample size is required to ensure reliability. Further research is required on data processing methods for the abutment, build orientation, and printing-layer thickness.

Evaluation of the gap in esthetic dental prostheses is necessary to reinforce the conclusions of this study. For the complete utilization of AM technology in the dental field, ongoing research should incorporate a fit analysis based on the results of this study.

## 5. Conclusions

Within the limitations of this in vitro study, the incisal, internal, and marginal gaps of the 3D-printed zirconia laminates were below the clinically acceptable range, indicating the potential of 3D printing as a new and effective method for producing laminates. The gap between the zirconia laminates produced using DLP printing and the SM method did not differ significantly between the manufacturing methods, except at the mesioincisal measurement point. Further research on different 3D printing methods and the development of more translucent 3D-printed ceramic materials is required.

## Figures and Tables

**Figure 1 biomimetics-09-00728-f001:**
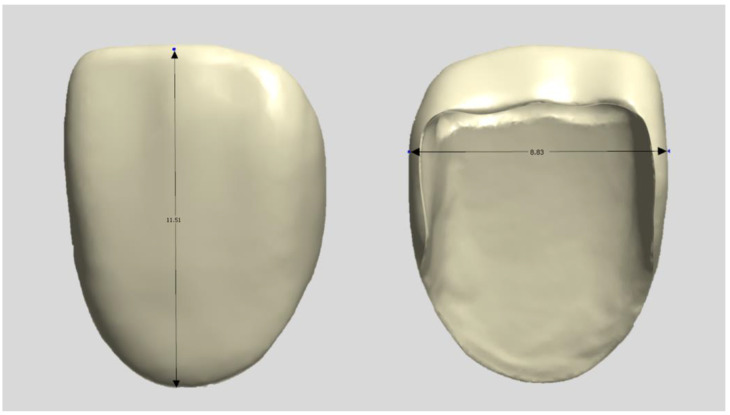
Design file of a specimen.

**Figure 2 biomimetics-09-00728-f002:**
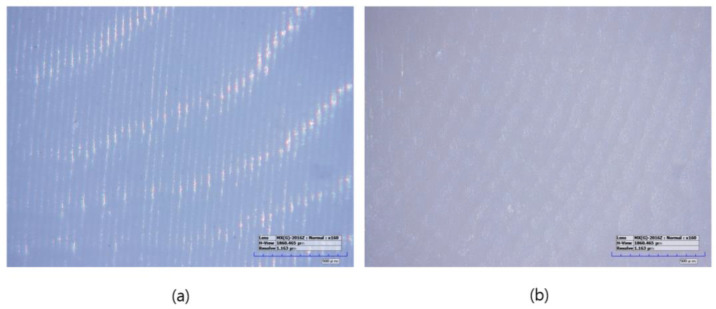
Microscopic images of sintered specimens (160× magnification): (**a**) LAM group, (**b**) LSM group.

**Figure 3 biomimetics-09-00728-f003:**
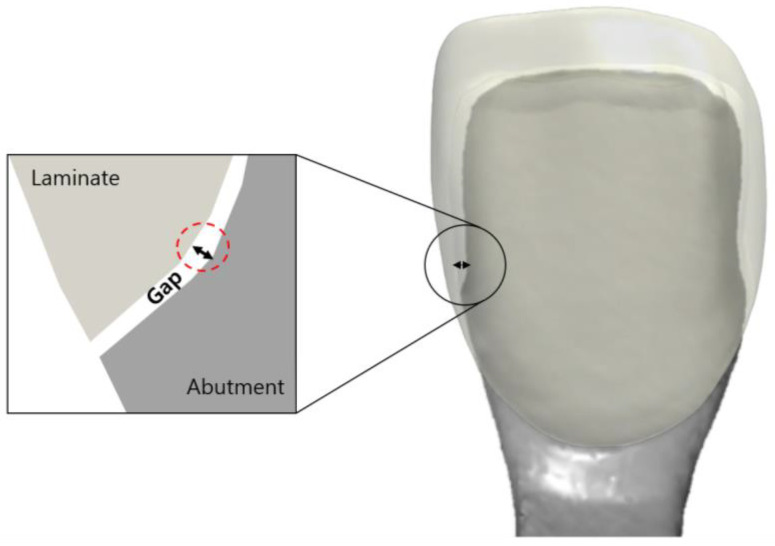
Schematic drawing of the gap between the inner surface of the laminate and the outer surface of the abutment tooth.

**Figure 4 biomimetics-09-00728-f004:**
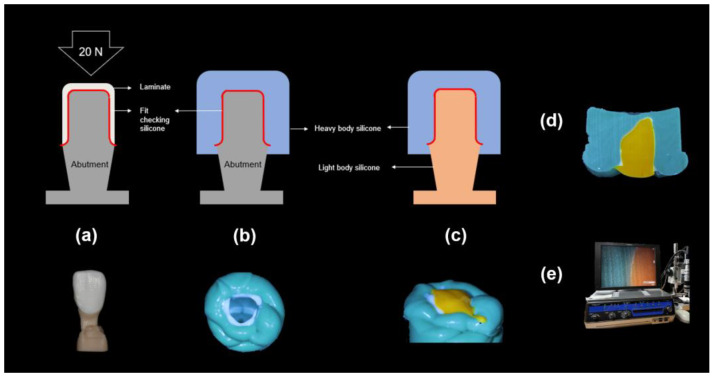
Process of the silicone replica method: (**a**): The abutment model filled with the fit-checking silicone and 20 N pressure, (**b**): Stabilization with the heavy body silicone, (**c**): Stabilization with the light body silicone, (**d**): Microscopic specimen, and (**e**) Digital microscope measurements.

**Figure 5 biomimetics-09-00728-f005:**
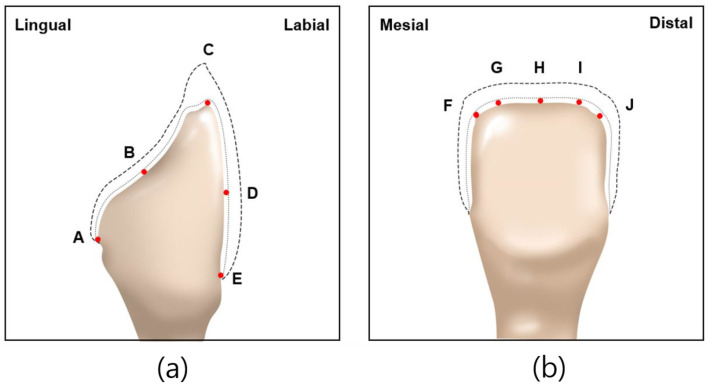
Ten measuring points selected on abutment in labiolingual and mesiodistal planes: (**a**) measuring points A, B, C, D, E of labiolingual plane; (**b**) measuring points F, G, H, I, J of mesiodistal plane.

**Figure 6 biomimetics-09-00728-f006:**
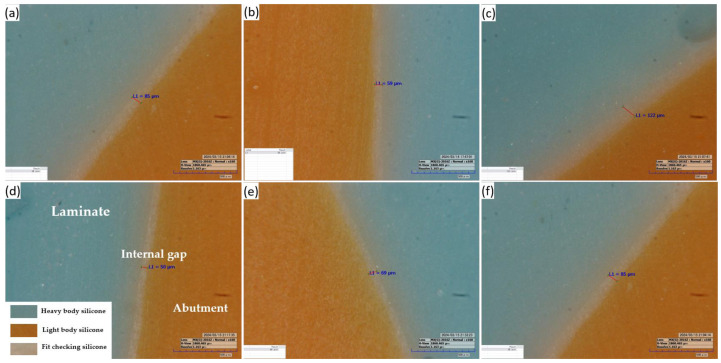
Representative image of the silicone replica method: (**a**,**b**,**d**,**e**) at labiolingual plane and (**c**,**f**) at mesiodistal plane (160× magnification).

**Figure 7 biomimetics-09-00728-f007:**
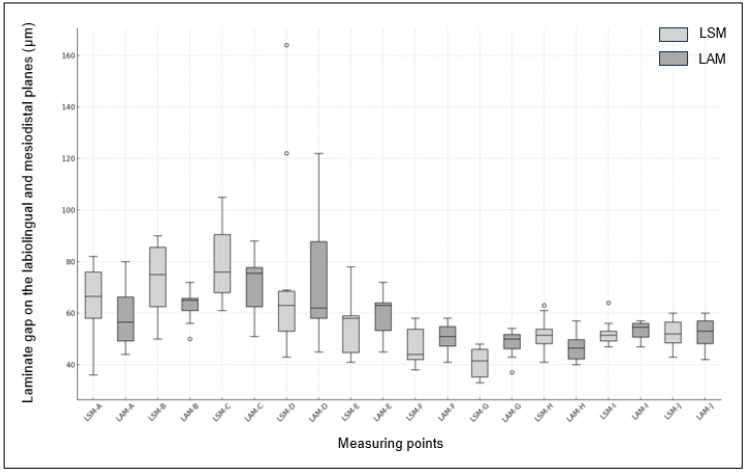
Box plot of fit between LSM and LAM Groups. (The *x*-axis represents the measuring points (A, C, D, E, F, G, and I) and the *y*-axis shows the measurement values.).

**Table 1 biomimetics-09-00728-t001:** Comparison of fit in labiolingual measurement points (A, B, C, D, and E) (LSM: Milling laminate group; LAM group: DLP printing laminate group).

Section	Group	Mean (Median)	SD	Max	Min	95% CI of Difference	*p*-Value
A	LSM	65.60 (66.5)	14.08	82	36	55.53–75.67	0.218
	LAM	59.1 (56.5)	11.88	80	44	50.6–67.6	
B	LSM	73.1 (75)	13.72	90	50	63.28–82.91	0.075
	LAM	63.1 (65)	6.27	72	50	58.61–67.59	
C	LSM	79.7 (76)	15.73	105	61	68.44–90.96	0.436
	LAM	70.9 (75.7)	12.74	88	51	61.78–80.02	
D	LSM	74.5 (63)	38.49	164	43	46.97–102.03	0.853
	LAM	72.6 (62)	23.16	122	45	56.03–89.17	
E	LSM	56.2 (58)	13.02	78	41	46.89–65.51	0.393
	LAM	59.5 (63)	8.73	72	45	53.25–65.75	

**Table 2 biomimetics-09-00728-t002:** Comparison of fit in mesiodistal fit measurement points (F, G, H, I, and J) (LSM: Milling laminate group; LAM group: DLP printing laminate group).

Section	Group	Mean (Median)	SD	Max	Min	95% CI of Difference	*p*-Value
F	LSM	47.20 (44)	7.50	58	38	41.84–52.56	0.280
	LAM	50.30 (51)	5.70	58	41	46.22–54.38	
G	LSM	40.70 (41.5)	5.93	48	33	36.46–44.94	0.004
	LAM	48.60 (50)	5.38	54	37	44.75–52.45	
H	LSM	51.70 (51.5)	6.68	63	41	46.92–56.48	0.143
	LAM	47.10 (46.5)	5.97	57	40	42.83–51.37	
I	LSM	52.30 (51.5)	4.90	64	47	48.79–55.81	0.247
	LAM	53.50 (54.5)	3.44	57	47	51.04–55.96	
J	LSM	47.20 (44)	7.50	58	38	41.84–52.56	0.971
	LAM	50.30 (51)	5.70	58	41	46.22–54.38	

## Data Availability

The raw data supporting the conclusions of this article will be made available by the authors upon request.

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
