# Peer review of "A Comparison of Internal, Marginal, and Incisal Gaps in Zirconia Laminates Fabricated Using Subtractive Manufacturing and 3D Printing Methods"

_biomimetics, 2024, doi:10.3390/biomimetics9120728_

Round 1

Reviewer 1 Report

Comments and Suggestions for Authors

This study discusses the DPL printing method. This topic may be of interest to the reader. The following minor corrections would be appropriate.

Abstract: A conclusion sentence more relevant to the findings should be written.

Introduction: Examples of materials that have optically similar properties to the natural teeth mentioned in lines 34-35 should be given.

The purpose sentence in lines 91-92 should provide more detail.

Materials and methods: Reference should be made to the production procedures of the samples.

The resolution of Figure 1 should be increased. More detailed and larger images should be included.

Reference to testing procedures should be added.

Results: Well written.

Discussion: The limitations of the study are written. However, they should be rewritten more compactly at the end of this section.

Conclusion: Well written.

References: Many references are older than 5 years. If possible, they should be replaced with new ones.

Comments on the Quality of English Language

The text is understandable. There are no significant errors in terms of language.

Author Response

Abstract: A conclusion sentence more relevant to the findings should be written.

Answer : I would like to thank the reviewer for having suggested this detailed point. A conclusion (line 22-24) have been thoroughly revised according to the reviewer.

Introduction: 

Examples of materials that have optically similar properties to the natural teeth mentioned in lines 34-35 should be given.

Answer : Thank you for pointing this out. I revised lines 36-37 to include "glass-based lithium disilicate ceramics, alumina, and zirconia," which have optical properties similar to natural teeth. I also replaced reference [8] with a new reference that is less than five years old. (line 36-38)

The purpose sentence in lines 91-92 should provide more detail.

Answer : Thank you for pointing this out. I agree with this comment. Therefore, I added more details to lines 98-99, noting that only studies on the fit of laminates manufactured exclusively through SLA 3D printing have been conducted.

Materials and methods: 

Reference should be made to the production procedures of the samples.

Thank you for pointing this out. Therefore, I added reference [43] to the production of the samples.

The resolution of Figure 1 should be increased. More detailed and larger images should be included.

Answer : Considering the reviewer’s point, more images were included and the resolution was increased.

Reference to testing procedures should be added.

Thank you for pointing this out. Therefore, I added reference [44,45] to testing procedures

Results: Well written.

Answer : Thank you for the revision.

Discussion: The limitations of the study are written. However, they should be rewritten more compactly at the end of this section.

Answer : Thank you for the revision.

Conclusion: Well written.

Answer : Thank you for the revision.

References: Many references are older than 5 years. If possible, they should be replaced with new ones.

Answer : Considering the reviewer’s point, i revised reference [1], [11], [12], [15], [16], [18], [19], [33], [36], [43], [44], [45].

Reviewer 2 Report

Comments and Suggestions for Authors

The article "Comparison of internal, marginal and incisal gaps in zirconia laminates fabricated by subtractive manufacturing and 3D printing methods" may be published in the journal Biomimetics after revision.

Remarks

Introduction

1. The first sentence of the Introduction contains the phrase "Currently, the purpose of dental prosthetic treatment has evolved beyond merely restoring caries...". However, there is no evidence to support this. Perhaps, the study https://doi.org/10.3390/polym12051176 should be considered.

 2. The scientific novelty of the work should be indicated, since DLP technology for processing ceramics is well known (see, for example, the work https://doi.org/10.1016/j.jeurceramsoc.2020.05.079).

Materials and methods

3. Captions should be placed below the figures, not above them.

4. It is necessary to explain what the resin is in which yttrium oxide is dispersed (what binder was used).

Conclusions

5. It is necessary to indicate the prospects for further research in this area.

Author Response

Introduction

  1. The first sentence of the Introduction contains the phrase "Currently, the purpose of dental prosthetic treatment has evolved beyond merely restoring caries...". However, there is no evidence to support this. Perhaps, the study https://doi.org/10.3390/polym12051176 should be considered.

Answer : Agree. I , accordingly, revised the line 31-33.

  1. The scientific novelty of the work should be indicated, since DLP technology for processing ceramics is well known (see, for example, the work https://doi.org/10.1016/j.jeurceramsoc.2020.05.079).

Answer : Thank you for pointing this out. Therefore, I revised the introduction line 95-98 to indicate the scientific novelty.

Materials and methods

  1. Captions should be placed below the figures, not above them.

Answer : Thank you for pointing this. I revised all the figures.

  1. It is necessary to explain what the resin is in which yttrium oxide is dispersed (what binder was used).
    Answer : Thank you for the revision. I revised line 123-125 “The bath of the 3D printer was filled with a zirconia paste (ININI-CERA, Aon, Seoul, Ko-rea). The paste consists of 3-mol%-yttria-stabilized zirconia powder with a photopoly-merized resin serving as the binder.”.

Conclusions

  1. It is necessary to indicate the prospects for further research in this area.

Answer : Considering the reviewer’s point, I revised the prospects for further research in the conclusion. (;line 328-329)

Reviewer 3 Report

Comments and Suggestions for Authors

1. Please provide further explanation of what zirconia laminates produced by DLP printing and milling methods are, using graphic forms.

2. Please briefly explain how zirconia laminates are used to repair teeth. What is the basic shape of the zirconia laminates used in this method, and what are the special requirements for its thickness and other geometric dimensions?

3. Please briefly explain what the zirconia laminates produced by mechanical processing and those produced by DLP printing are like in terms of state. Can the author provide a graphic explanation?

4. What is the contraction rate and consistency of the material during the post-processing of DLP 3D printing?

5. What kind of raw material is used in the subtractive processing, and what is its shape? What is the shrinkage rate during the post-processing?

6. A gap in a prosthesis is the difference between the inner and outer surfaces of an abutment tooth. Please provide a graphic explanation of the principle of how the gap is formed.

7. Please provide a graphic explanation of the steps and basic principles of the silicone replica method.

Comments on the Quality of English Language

The English could be improved to more clearly express the research.

Author Response

1. Please provide further explanation of what zirconia laminates produced by DLP printing and milling methods are, using graphic forms.

Answer :  I would like to thank the reviewer for having suggested this detailed point. I added (Figure 1. Design file of a specimen) to provide further explanations.

2. Please briefly explain how zirconia laminates are used to repair teeth. What is the basic shape of the zirconia laminates used in this method, and what are the special requirements for its thickness and other geometric dimensions?

Answer : Thank you for pointing this out. I revised Figure 1 to explain details of laminate. (Thickness 0.1~0.3mm, length 11.51 and width 8.71)

3. Please briefly explain what the zirconia laminates produced by mechanical processing and those produced by DLP printing are like in terms of state. Can the author provide a graphic explanation?

Answer : Thank you for pointing this out. I revised Figure 2 to provide a microscopic image.

4. What is the contraction rate and consistency of the material during the post-processing of DLP 3D printing?

Answer :  As for the shrinkage rate that occurs after completing the debinding and sintering processes of the DLP 3D printed zirconia, previous research has shown that the average shrinkage of the specimen in the x, y, and z directions is 22.11%, 22.51%, and 25.31%, respectively, as a result of testing on bar-shaped specimens.

Previous study : Mohammed, M. K., Alahmari, A., Alkhalefah, H., & Abidi, M. H. (2024). Evaluation of zirconia ceramics fabricated through DLP 3d printing process for dental applications. Heliyon10(17).

5. What kind of raw material is used in the subtractive processing, and what is its shape? What is the shrinkage rate during the post-processing?

Answer : Thank you for pointing this out. I added more details to line 116, noting that the pre-sintered 3 mol% zirconia disk block was used in the subtractive processing. Also, post-processing shrinkage is usually 20% to 25%, but detailed figures are not disclosed by the material company.

6. A gap in a prosthesis is the difference between the inner and outer surfaces of an abutment tooth. Please provide a graphic explanation of the principle of how the gap is formed.

Answer :  I would like to thank the reviewer for having suggested this detailed point. I added (Figure 3. Schematic drawing of the gap between inner surface of the laminate and outer surface of the abutment tooth).

7. Please provide a graphic explanation of the steps and basic principles of the silicone replica method.

Answer : : Considering the reviewer’s point, (Figure 4. Schematic of the silicone replica method : (a): The abutment model filled with the fit checking sili-cone and 20 N pressure, (b): Stabilization with the heavy body silicone, (c): Stabilization with the light body silicone) was added.

Round 2

Reviewer 2 Report

Comments and Suggestions for Authors

The authors have taken into account all the recommendations and made the appropriate changes to the manuscript. I believe the manuscript is ready for publication.

Author Response

Thank you for the revision. 

Reviewer 3 Report

Comments and Suggestions for Authors

The author's research has certain scientific significance for 3D printing of dental patches, but the arguments presented are mostly expressed in language. Please put the hardware equipment such as printers and machine tools into supplementary materials, and show the process of replica method with photos appropriately to increase the readability of the article.

Comments on the Quality of English Language

The English is fine.

Author Response

Please put the hardware equipment such as printers and machine tools into supplementary materials, and show the process of replica method with photos appropriately to increase the readability of the article.

 Answer :

Thank you for pointing this out. The details of hardware equipment used in this study are prepared are attached as a word file.

 I would like to thank the reviewer for having suggested this detailed point. The process of replica method was revised with photos in Figure 4. (line 151-154)